# Sp1 mediated the inhibitory effect of glutamate on pulmonary surfactant synthesis

**Xiao-Hong Li[1‡], Jie-Jun Fu[2], Xiao-Juan Shi[2], Yun-Na Zhang[2], Min Shao[2], Shao-Jie Yue[3], Chen Li[2,4]\*, Zi-Qiang Luo[2,5]\***

**1** Department of Pathology, The Second Xiangya Hospital, Central South University, Changsha, Hunan, China, **2** Department of Physiology, Xiangya School of Medicine, Central South University, Changsha, Hunan, China, **3** Department of Pediatrics, Xiangya Hospital, Central South University, Changsha, China, **4** Department of Physiology, Changzhi Medical College, Changzhi, Shanxi, China, **5** Hunan Key Laboratory of Organ Fibrosis, Central South University, Changsha, Hunan, China

‡ XHL share first authorship on this work.
* luoziqiang@csu.edu.cn (ZQL); Chen.physiology@outlook.com (CL)

**Data Availability Statement:** All relevant data are within the paper and its Supporting Information files.

**Funding:** This research was supported by the National Natural Science Foundations of China (No. 81900070 for Li XH, 81870059 for Luo ZQ,

## Abstract

### Background

Studies have shown that the release of endogenous glutamate (Glu) participates in lung injury by activating N-methyl-D-aspartate receptor (NMDAR), but the mechanism is still unclear. This study was to investigate the effects and related mechanisms of Glu on the lipid synthesis of pulmonary surfactant (PS) in isolated rat lung tissues.

### Methods

The cultured lung tissues of adult SD rats were treated with Glu. The amount of [³H]-choline incorporation into phosphatidylcholine (PC) was detected. RT-PCR and Western blot were used to detect the changes of mRNA and protein expression of cytidine triphosphate: phosphocholine cytidylyltransferase alpha (CCTα), a key regulatory enzyme in PC biosynthesis. Western blot was used to detect the expression of NMDAR1, which is a functional subunit of NMDAR. Specific protein 1 (Sp1) expression plasmids were used. After transfected with Sp1 expression plasmids, the mRNA and protein levels of CCTα were detected by RT-PCR and Western blot in A549 cells. After treated with NMDA and MK-801, the mRNA and protein levels of Sp1 were detected by RT-PCR and Western blot in A549 cells.

### Results

Glu decreased the incorporation of [³H]-choline into PC in a concentration- and time- dependent manner. Glu treatment significantly reduced the mRNA and protein levels of CCTα in lungs. Glu treatment up-regulated NMDAR1 protein expression, and the NMDAR blocker MK-801 could partially reverse the reduction of [³H]-choline incorporation induced by Glu ($10^{-4}$ mol/L) in lungs. After transfected with Sp1 plasmid for 30 h, the mRNA and protein expression levels of CCTα were increased and the protein expression of Sp1 was also up-regulated. After A549 cells were treated with NMDA, the level of Sp1 mRNA did not change significantly, but the expression of nucleus protein in Sp1 was significantly decreased, while

82070068 for Luo ZQ) and the Natural Science Foundation of Hunan Province (No. 2020JJ5813 for Li XH). Li XH and Luo ZQ had roles in study design, data collection and analysis, decision to publish, or preparation of the manuscript.

**Competing interests:** The authors have declared that no competing interests exist.

the expression of cytoplasmic protein was significantly increased. However, MK-801could reverse these changes.

## Conclusions

Glu reduced the biosynthesis of the main lipid PC in PS and inhibited CCTα expression by activating NMDAR, which were mediated by the inhibition of the nuclear translocation of Sp1 and the promoter activity of CCTα. In conclusion, NMDAR-mediated Glu toxicity leading to impaired PS synthesis may be a potential pathogenesis of lung injury.

## 1. Introduction

Glutamate (Glu) is the most abundant free amino acid in our brain and also the main excitatory neurotransmitter. It is a multifunctional amino acid and involves in memory formation, brain function etc. Extracellular excessive Glu can activate its *N*-methyl-*D*-aspartate (NMDA) receptor (NMDAR) and cause Glu toxicity [1]. It is characterized by damage to cellular components, ultimately leading to cell dysfunction and even death in neurons [1]. So far, there are seven different NMDAR subunits, which can be divided into three subfamilies according to sequence homology: NMDAR1 subunit, four different NMDAR2 subunits encoded by four different genes (2A, 2B, 2C, and 2D), and a pair of NMDAR3 subunits from two independent genes (3A and 3B) [2]. Among them, NMDAR1 is the basic subunit of NMDAR and the functional subunit of different NMDAR complexes, which plays an important role in the realization of the function of NMDAR ion channel [3]. In recent years, the role of NMDAR-mediated Glu toxicity in peripheral organizations has attracted more attentions. Studies have shown that NMDAR also exits in islets [4], lungs [5], alveolar macrophages [6], and bone marrow-derived mesenchymal stem cells [7].

Acute lung injury (ALI) is a critical illness syndrome with high morbidity and mortality. Its pathogenesis has not yet been fully elucidated. Studies have shown that NMDAR-mediated Glu toxicity is involved in the process of lung injury [5]. NMDA is a synthetic agonist that selectively activates NMDAR, while MK-801 and memantine are NMDAR blockers [8]. One study reported for the first time that NMDA could cause acute lung injury, and MK-801 could reverse this injury [9]. Our previous studies confirmed that hyperoxia or bleomycin challenge led to an increase in the release of endogenous Glu, while MK801 and memantine could effectively attenuate the ALI induced by hyperoxia [10] or bleomycin [11]. Endogenous Glu release mediated hyperoxia-induced lung injury in neonatal rats through NMDAR activation [12]. These studies reveal that NMDAR-mediated Glu toxicity may be a potential factor in the pathogenesis of ALI, but the mechanism is still unclear.

Pulmonary surfactant (PS) is a phospholipid protein mixture synthesized and secreted by alveolar type II (AT-II) cells. It is distributed on the surface of the molecular layer of alveolar fluid and has the functions of reducing alveolar surface tension and regulating the defense and protection mechanism in the lung [13]. PS is an indispensable bioactive substance to maintain normal respiratory function. The damage of AT-II cells or PS plays an important role in the pathogenesis of various lung diseases, including ALI [14]. PS is a complex mixture composed of 90% lipids and 10% proteins, in which phosphatidylcholine (PC) is the main component of lipids and the active component of PS to reduce alveolar surface tension. CTP: phosphocholine cytidylyltransferase α (CCTα) is a key regulatory enzyme involved in PC biosynthesis and the reduced activity of CCTα is an important cause of PS synthesis disorders in pathological

situations [15]. We previously reported that NMDAR activation inhibited PC synthesis and CCTα expression [16, 17], but its mechanism is not clear.

Transcription factor specific protein 1 (Sp1) is widely expressed in various cells *in vivo*, which plays an important role in regulating many housekeeping genes, especially in determining the expression of genes lacking TATA box in the promoter, and participates in various physiological and pathological processes such as cell proliferation, differentiation, apoptosis, and tumorigenesis [18]. Research shows that the proximal promoter region of CCTα gene contains multiple Sp1 binding sites, indicating that Sp1 plays an important role in the transcriptional activation and regulation of CCTα gene [19]. This study was to observe the effect of Glu on PS lipid biosynthesis in isolated rat lung tissue and the effect of NMDAR activation on Sp1 expression in A549 cells, so as to explore the effect and mechanism of NMDAR-mediated Glu toxicity on PS biosynthesis.

## 2. Materials and methods

### 2.1. Ethics statement

The Ethics Committee of Central South University (approval No: 2020297, Changsha, China) approved the experiments, which were performed in accordance with the guidelines of National Institutes of Health (NIH) for the care and use of animals.

### 2.2. Experimental animals

Twelve adult female Wistar rats (180~220g) were purchased from JingDa Laboratory Animal Company (Changsha, China). The purchased rats were fed in the Experimental Animal Center with specific pathogen free (SPF) of Xiangya School of Medicine, Central South University (Changsha, China). They were maintained in 12-h light/dark cycles at 20˚C with free access to food and water in accordance with guidelines from the Research Animal Welfare Committee of Central South University. We observed the feeding and drinking conditions of rats, weighed, and replaced the padding every day. All rats had normal weight and were used in the experiment without death.

### 2.3. Lung tissues cultured without serum

On the day of taking lung tissues, we prepared reagents and instruments in advance, and then moved the rats from the feeding room to the operation room for euthanasia, which lasted about half an hour. Rats were anesthetized with pentobarbital sodium solution at a dose of 100~200 mg/kg i.p body weight to achieve euthanasia. After ensuring that the animals had no vital signs (no ups and downs in the chest, white eyelids, no visual responses, etc.), the rats were exposed the heart and lungs. After removal of the lung blood by pulmonary artery cannula with saline, the lungs were cut into pieces of approximately 1 mm$^3$ using sterile scissors. It took about half an hour from the determination of euthanasia to being made into lung pieces for a rat. According to the guidelines of the Research Animal Welfare Committee of Central South University, rat carcasses were treated in standard procedures.

After washed with precooled dulbecco's modified eagle's medium (DMEM) for three times, about 20~30 pieces of lung tissues were placed into the culture dishes, and added 1ml serum-free DMEM containing penicillin (100 U/ml), streptomycin (100 µg/ml), and L-arginine ($10^{-4}$ mol/L). The tissue pieces were partly in contact with the atmosphere and partly with the culture medium. The tissues were then cultured at 37˚C in a humidified atmosphere of 5% $CO_2$ and 95% air for 8 h.

According to experimental design, the lung tissues were divided into experimental group and control group. The lung tissues were treated with different concentrations ($10^{-5} \sim 10^{-2}$ mol/L) of Glu for 16 h or treated with $5 \times 10^{-4}$ mol/L Glu for different times (2 h, 4 h, 8 h, 16 h, 24 h) for detection of incorporation of [$^3$H]-choline into PC. After treated with $10^{-4}$ mol/L Glu and $1.5 \times 10^{-6}$ mol/L MK-801 for 16 h, the lung tissues were used for detection of incorporation of [$^3$H]-choline into PC. After incubated with $10^{-4}$ mol/L Glu for 8 h or 16 h, the cultured lungs were collected for measuring the expression levels of CCTα and NMDAR1 respectively.

## 2.4. Cell culture

The human lung adenocarcinoma cell line A549 cells were obtained from the central laboratory of Xiangya School of Medicine (Changsha, China). A549 cells were cultured in DMEM medium (Gibco, USA) supplemented with 10% fetal bovine serum (FBS).

## 2.5. Detection of incorporation of [$^3$H]-choline into PC

For evaluating the rate of phospholipid synthesis, cultured lung explants were exposed to [methyl-$^3$H] choline chloride (0.2 μCi/ml) in defined medium. At the end of the incorporation period, the radioactive medium was removed and cells were rinsed three times with phosphate-buffered saline before made-up 10% lung homogenate. The lung tissue homogenate had its lipids extracted by chloroform/methanol 2:1 (vol/vol). Dried by blow with nitrogen gas, and redissolved in 200 μl chloroform. The liquid scintillation count (Beckman LS3801) was used to determine incorporation of [$^3$H]-choline into PC. Total proteins were measured by Lowry method and the results were shown in cpm/mg protein.

## 2.6. RT- PCR and RT-qPCR

Total RNA was extracted from whole lung tissues or A549 cells with TRIZOL reagent (Invitrogen) according the manufacturer's protocol. Total RNA (1 μg) was reverse transcribed into cDNA using a PrimeScript RT Reagent Kit (Thermo Scientifific) with gDNA Eraser following the manufacturer's instructions.

Real-time PCR (RT-PCR) was used for detection CCTα mRNA expression. The primers were used as following: CCTα (F: ATTGTCCGTGACTATGATGTG, R: CTTGGGACTGATGGC CTG); β-actin (F: TGGCTACAGCTTCACCACC, R: ACTCCTGCTTGCTGATCCAC). PCR reaction was done in the applied biosystems of ProFlex™ PCR system (Thermo Scientifific). The DNA signals were observed in the Molecular Imager ChemiDoc XRS System (Bio-Rad). Quantification of data and subsequent statistical analyses were performed with Image Lab analysis software.

Quantitative real-time PCR (RT-qPCR) was used for detection Sp1 mRNA expression. The primers were used as following: Sp1 (F: CAACTTGCAGCAGAATTGAGTC, R: TGTTCCTTTGA GGTAGGGGTAG); β-actin (F: TGACGTGGACATCCGCAAAG, R: CTGGAAGGTGGACAGCGA GG). The analyses of SYBR Premix Ex Taq II (Takara) were done in the real-time PCR detection system (CFX96 Touch, Bio-Rad). The relative expression of mRNA was determined by normalizing the expression of each gene to β-actin gene following the $2^{-\Delta\Delta CT}$ method. Each experiment was performed in duplicate and repeated three times.

## 2.7. Plasmid preparation and transfection

Sp1 expression plasmid pEVR2-Sp1 and pEVR2 empty vectors came from Garold S. Yost professor of the university of Uta [20]. The plasmid sequencings were correct from Guangzhou

Ruizheng Sound Equipment Co. Lt. A549 cells were transfected with different plasmids with TRANSfection 2000 reagent (Invitrogen) following the manufacturer's instructions.

## 2.8. Western blot

Cells or tissues were lysed in RIPA lysis buffer plus proteinase inhibitor cocktail (Roche Diagnostics) to extract the total proteins. The extraction of nuclear proteins and cytoplasmic proteins was carried out using a Nuclear Protein Extraction Kit (Beijing Solarbio). The proteins concentrations were determined with BCA protein assay kit. The proteins were separated by SDS-PAGE, then transferred into PVDF membranes and blocked with 5% non-fat dry milk for 2 h at room temperature. Blots were incubated with primary antibodies (NMDAR1, CCTα and Sp1) (1:1,000) (Abcam) at 4˚C overnight. The internal control antibodies for the cytoplasmic proteins are β-actin and GAPDH (1:10,000) (Abcam), and the internal control antibody for nuclear proteins is Lamin A+C (1:10,000) (Abcam). The membranes were washed for three times in TBST and incubated with peroxidase-conjugated secondary antibodies (1:20,000) (Sigma) for 1 h at room temperature, protein signals were visualized with enhanced chemiluminescence detection kit in the Molecular Imager ChemiDoc XRS System (Bio-Rad). Quantification of data and subsequent statistical analyses were performed with Image Lab analysis software.

## 2.9. Statistical analysis

Data were expressed as mean ± SD. Statistical analysis was performed using Image-Pro Plus 6.0 software. Multiple comparison of mean values was made by analysis of variance (ANOVA). Two-variable comparison was made by two-tailed $t$-test in both instances. The value $P<0.05$ was considered to be statistically significant.

## 3. Results

### 3.1. Glu inhibited the PC synthesis in cultured lung tissues by NMDAR activation

PC is the main component of PS lipid, and [3H]-choline incorporation into PC is a commonly used indicator to observe the changes in PC synthesis [21]. To observe the effect of Glu on PC synthesis, $5\times10^{-4}$ mol/L Glu was added to treat the lung tissues for different times. The results showed that the amount of [3H]-choline incorporation in the Glu treatment group was significantly lower than that of the control group at 8 h, 16 h, 24 h ($P<0.05$, $P<0.05$, $P<0.01$) (Fig 1A). Compared with the control group, the incorporation of [3H]-choline decreased in a concentration dependent manner ($P<0.05$, $P<0.01$, $P<0.01$) (Fig 1A) when the lung tissues were exposed to Glu of different concentrations ($10^{-6}\sim10^{-2}$ mol/L) for 16 h. Glu concentration was negatively correlated with the incorporation of [3H]-choline (r = -0.9740) (Fig 1B). The above results indicated that Glu inhibited PS lipid synthesis in a time-dependent and dose-dependent manner in lung tissues of adult rats *in vitro*.

To further explore the role of NMDAR in mediating the inhibition of PC synthesis caused by Glu treatment, we used the NMDAR blocker MK-801 in this study. There was a significant difference between the Glu+MK-801 group and the Glu treatment group ($P<0.01$) (Fig 1C). The results showed that NMDAR blocker could reverse the inhibition of [3H]-choline incorporation induced by Glu treatment, suggesting that the inhibitory effect of Glu on PC synthesis was achieved by activating NMDAR.

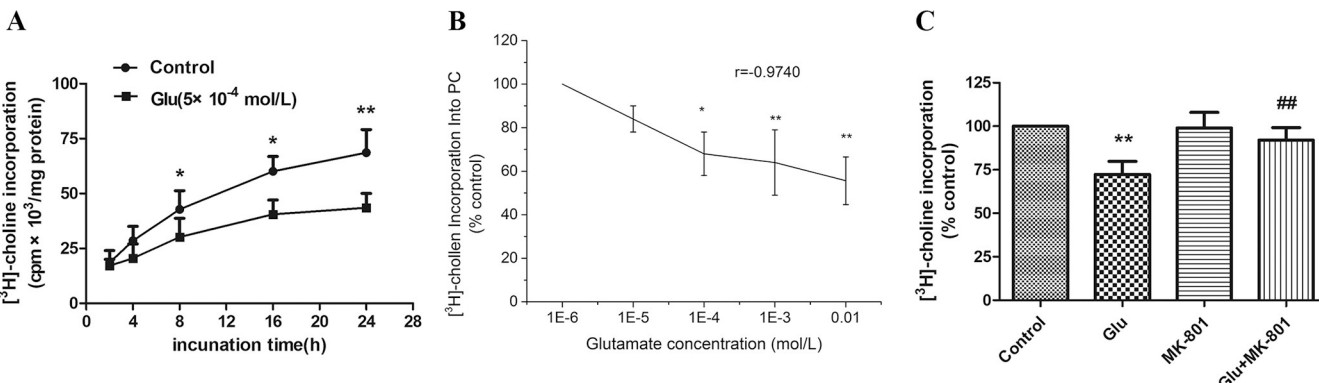

**Fig 1. Effects of Glu on PC synthesis in cultured lungs.** After different times and doses treatment, the cultured lungs were collected for measuring [$^3$H]-choline incorporation. (A) Results of [$^3$H]-choline incorporation into PC in cultured lung explants incubated with $5 \times 10^{-4}$ mol/L Glu for different times. (B) Results of [$^3$H]-choline incorporation into PC in cultured lung explants incubated with different doses of Glu for 16 h. (C) After treated with $10^{-4}$ mol/L Glu and $1.5 \times 10^{-6}$ mol/L MK-801 for 16 h, [$^3$H]-choline incorporation into PC were measured in cultured lung explants. Bars: mean ± SD. n = 5, *$P<0.05$; **$P<0.01$ vs. control, ##$P<0.01$ vs. Glu treatment group.

## 3.2. Glu decreased the mRNA and protein levels of CCTα in lung tissues

CCTα is a key enzyme for PC biosynthesis in mammalian cells. We then observed the effect of Glu on CCTα expression in cultured lungs. RT-PCR results showed the mRNA expression of CCTα in $10^{-4}$ mol/L Glu treatment group was significantly decreased compared with the control group ($P<0.01$) (Fig 2A and 2B). Western blot results showed the protein expression of CCTα in $10^{-4}$ mol/L Glu treatment group was significantly decreased compared with the control group ($P<0.01$) (Fig 2C and 2D). These results indicated that Glu decreased CCTα expression, and then inhibited PC synthesis.

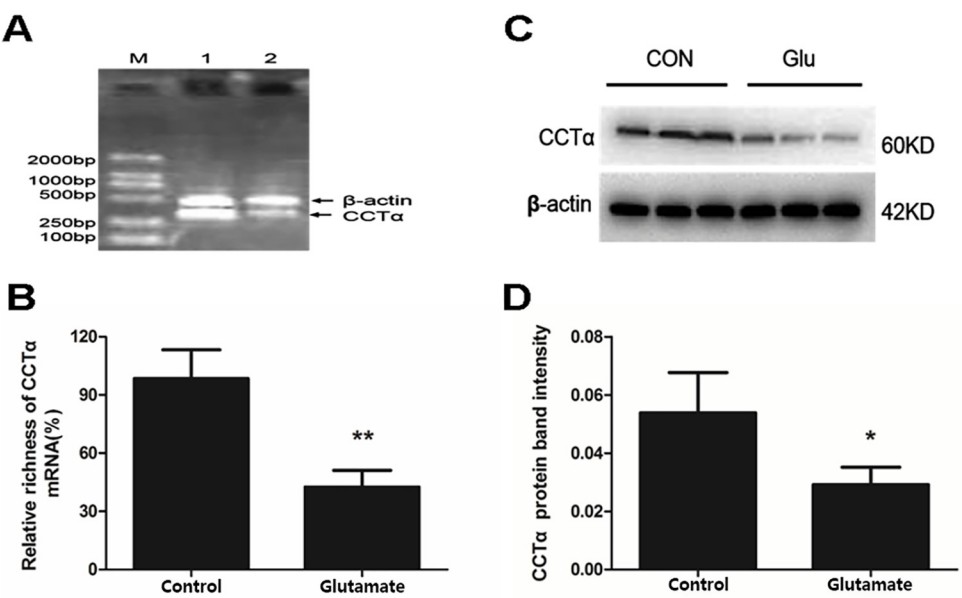

**Fig 2. Effects of Glu on CCTα mRNA and protein levels in cultured lungs.** After incubated with $10^{-4}$ mol/L Glu for 8 h, the cultured lungs were collected for measuring the levels of CCTα mRNA and protein expression by RT-PCR and Western blot. (A, B) Results of the RT-PCR and quantitative analysis of the relative gradation. (M: DNA marker line; 1: control line; 2: Glu treatment line). (C, D) Results of the Western blot and quantitative analysis of the relative gradation. Bars: mean ± SD. n = 4. *$P<0.05$, **$P<0.01$ vs. control.

### 3.3. Glu up-regulated the expression of NMDAR in lung tissues

NMDARs are heteromeric complexes incorporating different subunits within a repertoire of three subtypes: NMDAR1, NMDAR2 and NMDAR3. NMDAR1 is a functional subunit of NMDARs and the obligate member of all NMDARs. Western blot results showed there was NMDAR1 expression in the lungs and the expression level in the Glu treatment group was increased compared with the control group ($P<0.05$) (Fig 3), suggesting that Glu up-regulated the expression of NMDAR in lungs.

### 3.4. Sp1 upregulated the expression of CCTα

The above studies suggested that Glu decreased CCTα expression and then inhibited PC synthesis by activating NMDAR at the tissue level. Next, we further studied the mechanism of NMDAR-mediated Glu toxicity in reducing CCTα expression and PC synthesis at the cellular level. Specificity protein 1 (Sp1) is a key transcription factor and plays an important role in the transcriptional activation of CCTα-promoter reporter construct [19]. Activation of CCTα expression during the S phase of the cell cycle is mediated by Sp1 [22]. To observe the effect of Sp1 on CCTα expression, we used the pEVR2-Sp1 expression vector (Sp1 eukaryotic expression plasmid) to transfect into A549 cells and detected CCTα expression. The results showed that PEVR2-Sp1 increased the mRNA and protein expression levels of CCTα ($P<0.01$) (Fig 4A–4C) and the protein expression of Sp1 ($P<0.05$) (Fig 4D and 4E) in A549 cells. These results indicated that Sp1 promoted the expression of CCTα and played an important role in its stable expression.

### 3.5. Effects of NMDAR activation on the expression and nuclear translocation of Sp1 in A549 cells

Human cell line A549 cells have been widely used as the model of AT-II cells *in vitro*. NMDAR had also been reported to be expressed in A549 cells [23]. In cell experiments, we directly treated A549 cells with NMDA, an NMDAR-specific agonist. To observe the effect of NMDA on the level of Sp1 mRNA and protein in A549 cells, 300 μM NMDA was used to treat the cells for 1 h, 3 h, 5 h and 8 h respectively. RT-PCR results showed that the mRNA levels of Sp1 did not change significantly ($P>0.05$) (Fig 5). However, western blot results showed that the protein levels of Sp1 began to decrease at 5 h ($P<0.05$) in a time dependent manner, reaching the lowest point at 8 h ($P<0.01$) in the nuclear extract of A549 cells (Fig 6A and 6B). Moreover, after treated with different concentrations of NMDA for 8 h, the protein levels of Sp1 in

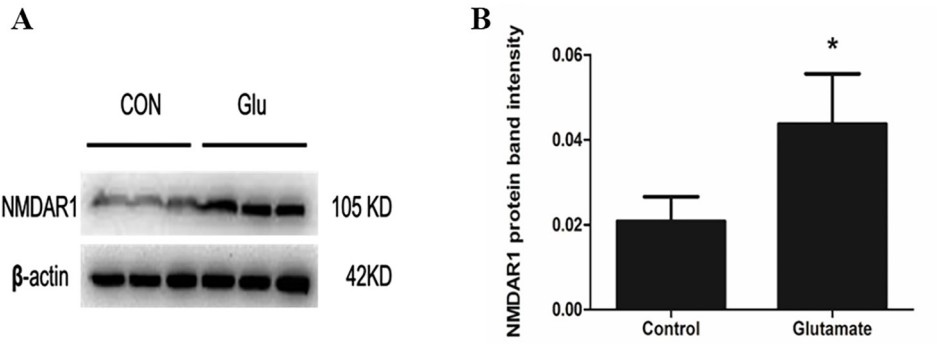

**Fig 3. Effects of Glu on NMDAR1 expression in cultured lungs.** After incubated with $10^{-4}$ mol/L Glu for 16 h, the cultured lungs were collected for measuring NMDAR1 expression by Western blot. (A) Results of the Western blot. (B) Quantitative analysis of the relative gradation. Bars: mean ± SD. n = 3. *$P<0.05$ vs. control.

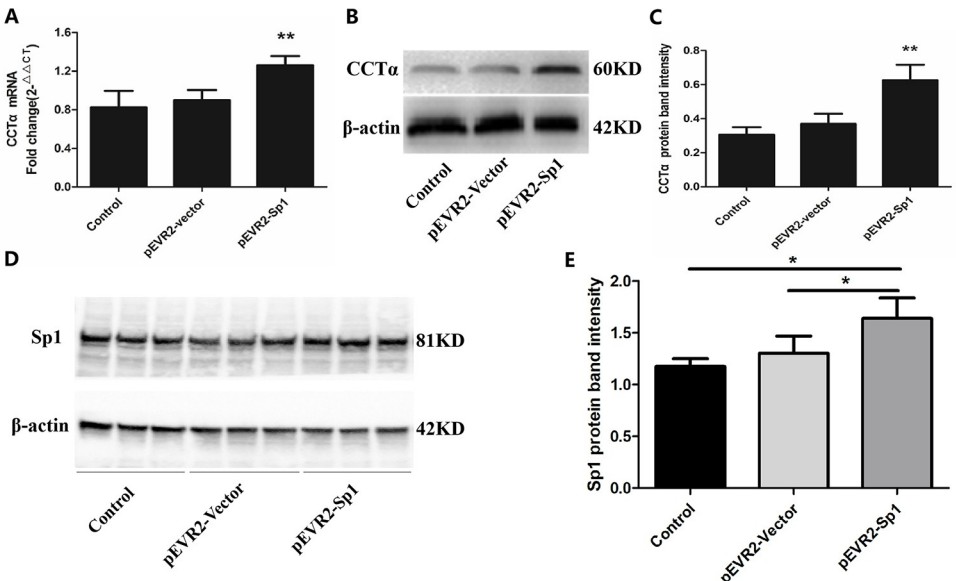

**Fig 4. Effects of Sp1 on the mRNA and protein expression of CCTα in A549 cells.** The mRNA and protein expression of CCTα was detected by RT-qPCR and Western blot in A549 cells after transfected the expression plasmids pEVR2-Sp1 and corresponding empty vectors pEVR2. (A) Results of the RT-qPCR. (B) Results of the Western blot. (C) Quantitative analysis of the relative gradation. Bars: mean ± SD. n = 3, **P<0.01 vs. Control. (D) Results of the Western blot. (E) Quantitative analysis of the relative gradation. Bars: mean ± SD. n = 3, *P<0.05.

nuclear protein were decreased in a concentration dependent manner (*P*<0.05, *P*<0.01) (Fig 6C and 6D).

Further, we performed the rescue experiments using the NMDAR blocker (MK-801). A549 cells were treated with 300 μM NMDA for 8 h simultaneously with 50 μM MK-801, and then

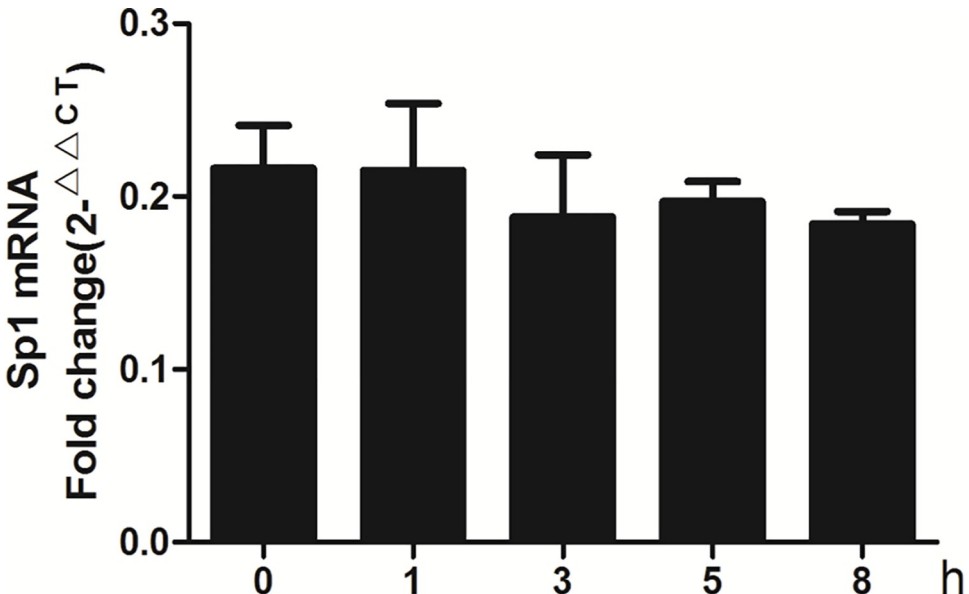

**Fig 5. Effects of NMDA on the mRNA expression of Sp1 in A549 cells.** 300 μM NMDA was used to treat A549 cells for indicated times (0 h, 1 h, 3 h, 5 h, 8 h), the mRNA levels of Sp1 were determined by RT-PCR. Bars: mean ± SD. n = 3, *P*>0.05 vs. Control.

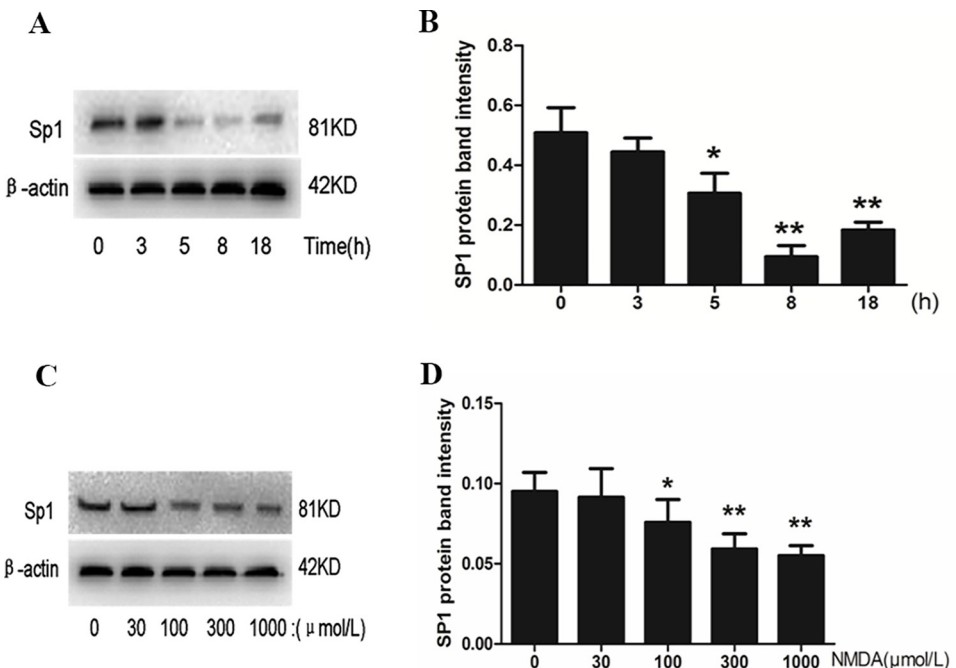

**Fig 6. Effects of NMDA on the nucleoprotein expression of Sp1 in A549 cells.** Protein levels of Sp1 induced by NMDA (300 μM) for indicated time were determined by Western blot (A, B). NMDA (30~1000 μM) indicated doses for 8 h were determined by Western blot (C, D). (A, C) Results of the Western blot. (B, D) Quantitative analysis of the relative gradation. Bars: mean ± SD, n = 3, *P<0.05, **P<0.01 vs. control.

the nuclear and cytoplasmic proteins were extracted. The expression of Sp1 was increased in the cytoplasmic proteins and decreased in the nuclear proteins after treated with 300 μM NMDA (P<0.05, P<0.01) (Fig 7A–7D). However, 50 μM MK-801 could reverse the increase of Sp1 expression in the cytoplasmic proteins and the decrease in the nuclear proteins induced by NMDA (P<0.05, P<0.01) (Fig 7A–7D). These results further suggested that NMDAR activation inhibited the nuclear translocation of Sp1. The above results indicated that NMDAR activation inhibited the nuclear translocation of Sp1 in A549 cells, suggesting that the decrease of CCTα induced by NMDAR activation was related to the inhibition of Sp1 nuclear translocation.

## 4. Discussion

Acute respiratory distress syndrome (ARDS) and its milder form acute lung injury (ALI) are the common clinical critical diseases, which are manifested by decreased lung compliance, severe hypoxemia, and bilateral pulmonary infiltrates. ALI/ARDS is characterized by diffuse alveolar injury, lung edema formation, neutrophil-derived inflammation, and pulmonary surfactant (PS) dysfunction [24]. PS is a lipoprotein complex that lines the alveoli and decreases the surface tension to prevent lung atelectasis, and plays an important role in maintaining normal pulmonary function. The endogenous PS deficiency contributes to the lung dysfunction associated with ALI/ARDS [25]. A large number of *in vitro* data and *in vivo* animal studies have demonstrated that PS not only maintains bronchiolar patency during biophysical functions, but also protects of the lungs from the injury and infection of inhaled particles and microorganisms [25]. The exogenous surfactant plays an important role in the treatment of ALI/ARDS [26].

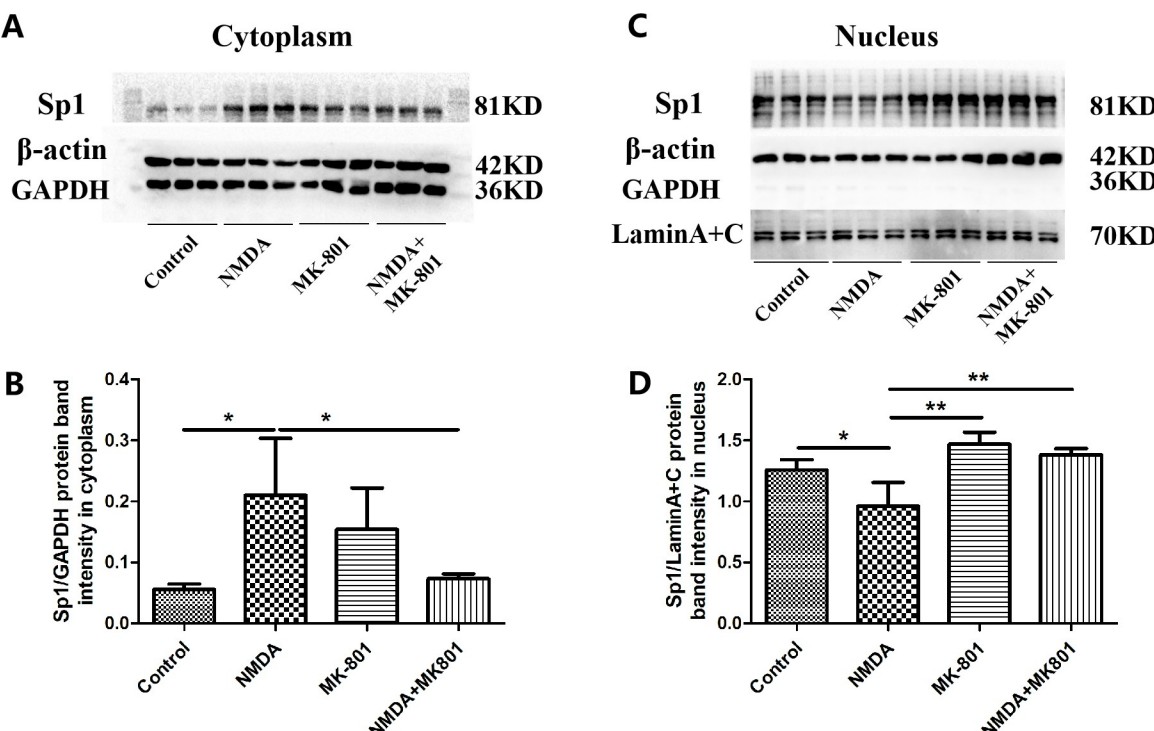

**Fig 7. Effects of NMDA on Sp1 nuclear translocation in A549 cells.** After treated with 300 μM NMDA and 50 μM MK-801 for 8 h, the protein expressions of Sp1 in cytoplasm and nucleus were detected by Western blot in A549 cells (A-D). (A, C) Results of the Western blot. (B, D) Quantitative analysis of the relative gradation. Bars: mean ± SD, n = 3, *P<0.05, **P<0.01.

*N*-methyl-*D*-aspartate receptor (NMDAR) is an important glutamate (Glu) receptor and ion channel, which is widely expressed in the central nervous system. NMDAR activation is a key mechanism of Glu toxicity and NMDAR-mediated Glu toxicity involves in different pathological changes [2]. Studies have shown that pretreatment with memantine can significantly reduce myocardial necrosis and cardiac fibrosis in rats with heart failure [27]. The previous study of our group also has showed that memantine can alleviate the pulmonary inflammation in mice with bleomycin-induced ALI [11]. The endogenous Glu release of bone marrow cells in BLM-injured mice is increased, while the anti-fibrotic ability of BM-MSCs pretreated with NMDA is decreased [7]. The above studies suggest that Glu toxicity mediated by NMDAR activation also plays an important role in ALI. In this study, the expression of NMDAR1 (a functional subunit of NMDAR) exists in lung tissue, and Glu up-regulates the protein expression of NMDAR1. Moreover, Glu inhibits the synthesis of phosphatidylcholine (PC), the main component of PS lipids, in a dose-dependent and time-dependent manner, while MK-801 can reverse the effect of Glu on the reduction of PC biosynthesis, suggesting that NMDAR activation mediated the reduction of PC synthesis induced by Glu. These studies suggest that NMDAR-mediated Glu toxicity plays an important role in the reduction of PS synthesis, which may be a potential mechanism for the lung injury in ALI/ARDS.

CTP: phosphocholine cyclically transfer (CCT) is a key regulatory enzyme in PC biosynthesis, which determines the rate of PC biosynthesis [28]. In mammals, this enzyme is encoded by two genes: Pcyt1a located on murine chromosome 16, that encodes the CCTα protein, and Pcyt1b located on the X chromosome which encodes the CCTβ2 and CCTβ3 proteins. There are three isoenzymes: CCTα, CCTβ1, and CCTβ2 [29]. CCTα protein is ubiquitously

expressed in nucleated cells and the main type of CCT in the lung. The mechanism of tumor necrosis factor (TNF) and reactive oxygen species (ROS) impairing PS biosynthesis is related to the decrease of CCTα activity in alveolar type II (AT-II) cells in the lung [30]. Our previous research found for the first time that NMDA down-regulated the mRNA and protein expression of CCTα and inhibited PC synthesis in human cell line A549 cells [16]. In this study, we also observed that Glu decreased the mRNA and protein expression of CCTα in isolated rat lung tissues. These results confirmed that Glu down-regulated CCTα via NMDAR activation, and then inhibiting PC biosynthesis. However, the mechanism of NMDAR activation in the down-regulation of CCTα expression remains to be further explored.

GC box specific binding protein specific protein (Sp) 1 (Sp1), a transcription factor widely expressed in mammalian genome, belongs to the Sp protein family. Sp1 has been proved to be involved in the transcription of a large number of genes, especially housekeeping (conservative) genes, tissue specific genes, and growth regulating genes [31]. Sp1 activates the transcription of many cellular genes that contain putative CG-rich Sp-binding sites in their promoters, thus enhance gene expression [32]. CCTα plays an important regulatory role in the induction of PC biosynthesis, which is an essential step during cell division. The distinguishing feature of CCTα is the presence of an N-terminal nuclear localization signal that directs the enzyme to the nucleus in many cultured cells [33]. Claudia Banchio et al. have reported that Sp1 binding to the CCTα proximal promoter is necessary to enhance transcription during the S phase for cellular mitosis [34, 35]. In this study, the dual-luciferase reporter assay system showed that CCTα promoter activity is increased with the increase of pEVR2-Sp1. Meanwhile, CCTα mRNA and protein levels were also increased. Therefore, these results are consistent: the activation of CCTα expression is mediated by the transcription factor Sp1.

To explore the mechanisms of CCTα down-regulation caused by NMDAR activation, we further observed the effect of NMDA on Sp1 in A549 cells, and found that NMDA did not affect the mRNA expression level of Sp1 (Fig 6), but could significantly reduce the level of nuclear Sp1 protein (Fig 7), which suggested that NMDAR activation could inhibit the translocation of Sp1 to the nucleus and down-regulate the content of nuclear Sp1 protein. It may be one of the important mechanisms of NMDAR activation inhibits the key enzyme CCTα of PC biosynthesis in AT-II cells. However, the changes in the level of nuclear Sp1 protein need further experimental research. In this study, we found that NMDA reduced the level of Sp1 protein in the nucleus by inhibiting its nuclear translocation, and weakened the effect of Sp1 on the activation of CCTα gene promoter. Our results suggested for the first time that the mechanism of Glu inhibiting PS synthesis is related to the effect of NMDAR activation on the down-regulation of CCTα expression via inhibiting Sp1 nuclear translocation.

## 5. Conclusion

In this study, we confirmed that Glu reduced lipid synthesis of PS by activating NMDAR in the lungs of adult rat. Furthermore, the mechanism was related to NMDA induced- inhibition of the nuclear translocation of Sp1, thereby reducing CCTα gene promoter activity. In summary, NMDAR-mediated Glu toxicity leading to impaired PS synthesis is one of the pathogeneses of lung injury.

## Supporting information

**S1 Raw images.**
(PDF)

## Acknowledgments

We thank that Garold S. Yost professor of the university of Utah kindly provided the expression plasmids of pEVR2-Sp1 and pEVR2 empty vectors.

## Author Contributions

**Conceptualization:** Zi-Qiang Luo.

**Data curation:** Xiao-Hong Li, Chen Li.

**Formal analysis:** Xiao-Hong Li, Jie-Jun Fu, Xiao-Juan Shi, Yun-Na Zhang, Min Shao, Chen Li.

**Funding acquisition:** Zi-Qiang Luo.

**Investigation:** Xiao-Hong Li.

**Resources:** Jie-Jun Fu, Shao-Jie Yue.

**Software:** Xiao-Juan Shi.

**Supervision:** Shao-Jie Yue, Zi-Qiang Luo.

**Validation:** Zi-Qiang Luo.

**Writing – original draft:** Xiao-Hong Li.

**Writing – review & editing:** Zi-Qiang Luo.

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
