## [Decision Letter · Decision Letter 0]

21 Feb 2023

PONE-D-22-33593Sp1 mediated the inhibitory effect of glutamate on pulmonary surfactant synthesisPLOS ONE

Dear Dr. Luo,

Thank you for submitting your manuscript to PLOS ONE. After careful consideration, we feel that it has merit but does not fully meet PLOS ONE’s publication criteria as it currently stands. Therefore, we invite you to submit a revised version of the manuscript that addresses the points raised during the review process.

We look forward to receiving your revised manuscript.

Kind regards,

Yi Cao

Academic Editor

PLOS ONE

Journal Requirements:

“This work was supported by the National Natural Science Foundations of China (No. 81900070,81870059,82070068) and the Natural Science Foundation of Hunan Province (No. 2020JJ5813).”           

Reviewers' comments:

Reviewer's Responses to Questions

**Comments to the Author**

1. Is the manuscript technically sound, and do the data support the conclusions?

Reviewer #1: Partly

2. Has the statistical analysis been performed appropriately and rigorously? 

Reviewer #1: Yes

3. Have the authors made all data underlying the findings in their manuscript fully available?

Reviewer #1: No

4. Is the manuscript presented in an intelligible fashion and written in standard English?

Reviewer #1: Yes

5. Review Comments to the Author

Reviewer #1: In this manuscript, Li and colleagues have tried to elucidate the mechanism of phosphatidylcholine biosynthesis under over-activated glutamate/NMDA signaling conditions. Most of what is described in the paper has been previously described in the literature, with the exception possibly being of cytoplasm translocation of Sp1 after NMDA treatment. However, the data from immunofluorescence staining cannot support their wording. Other more readily addressable issues are listed below.

1. (Line 194: Reporter plasmid of CCTα promoter PGL3-CCTα (-1893/+71) …) There is no data presented using this plasmid. Which species (human or rat) of the CCTα promoter sequence is used in reporter assay? Since the DNA sequence between humans and rats is different, the details of Sp1 binding sites within the CCTα promoter should be provided.

2. The data confirming Sp1 overexpression is lacking.

3. (Figure 5-7) The glutamate effects on Sp1 should be examined. Also, rescue experiments should be performed using the NMDAR blocker (MK-801).

4. (Line 285: These results indicated that NMDAR activation inhibited the nuclear translocation of Sp1 …) No data supported this wording unless the protein level of Sp1 is increased or maintained in the cytoplasm fractionation.

5. (Figure 7) The Sp1 immunofluorescence data of NMDA treatment is out of focus. The lens focus is on the mitotic cells. Sp1 is disassociated with mitosis DNA, as described in the previous literature. Also, the quality of image data should be improved and present higher magnification images.

6. Typo: Figure 2B, Glutamae -> Glutamate

6. PLOS authors have the option to publish the peer review history of their article (what does this mean?). If published, this will include your full peer review and any attached files.

Reviewer #1: No

---

## [Author Response · Author response to Decision Letter 0]

8 May 2023

Responses: Thank you for your reviews.

1.Reporter plasmid of CCTα core promoter PGL3-CCTα (-169/+71) and pRL-SV40 empty vector with luciferase labeled promoter were from Mallampalli R.K. professor of the university of Iowa. The murine CCTα core promoter fragment (-169/+71) was generated by PCR using the following primers: 5’-TTGTGTGTTTTCACCCCTTATG-3’ (left) and 5’-TCAACTCCTCCAGGCTCCGGT-3’ (right) as described in literature. The core promoter of CCTα gene was localized to a region between -169 and +71 bp, which exhibited strong basal activity comparable with a positive control that contains the simian virus 40 promoter. (Ryan AJ, Fisher K, Thomas CP, Mallampalli RK. Transcriptional repression of the CTP:phosphocholine cytidylyltransferase gene by sphingosine. Biochem J. 2004 Sep 1;382(Pt 2):741-50. doi: 10.1042/BJ20040105. PMID: 15139854; PMCID: PMC1133833.); (Mallampalli RK, Ryan AJ, Carroll JL, Osborne TF, Thomas CP. Lipid deprivation increases surfactant phosphatidylcholine synthesis via a sterol-sensitive regulatory element within the CTP:phosphocholine cytidylyltransferase promoter. Biochem J. 2002 Feb 15;362(Pt 1):81-8. doi: 10.1042/0264-6021:3620081. PMID: 11829742; PMCID: PMC1222362.)

 We conducted cell experiments using these plasmids in A549 cells and did obtain such result. We did not consider the issue of species at that time, but it was indeed a very thought-provoking issue. Therefore, we have decided to remove this result from the article to avoid unnecessary misunderstandings, but this does not affect our conclusion. As the reviewer pointed out, the impact of NMDAR activation on Sp1 nuclear translocation is the novelty of our study. 

Sp1 is part of a family of approximately 20 proteins that is characterized by having a DNA-binding domain that has 3 Krüppel-like zinc fingers (KLF). Studies have reinforced the central role for Sp1 in activating and promoting the transcription of CCTα gene. (Sugimoto H, Banchio C, Vance DE. Transcriptional regulation of phosphatidylcholine biosynthesis. Prog Lipid Res. 2008 May;47(3):204-20. doi: 10.1016/j.plipres.2008.01.002. Epub 2008 Feb 6. PMID: 18295604.)

Banchio et al. demonstrated that increased binding of the transcription factor Sp1 to the proximal promoter of CCTα is responsible for increased transcription during the S phase. The Sp1 binding element present in position -67/-62 is essential for activation, and the Sp1 site in position -31/-9 is required to enhance transcription. (Banchio C, Schang LM, Vance DE. Activation of CTP:phosphocholine cytidylyltransferase alpha expression during the S phase of the cell cycle is mediated by the transcription factor Sp1. J Biol Chem. 2003 Aug 22;278(34):32457-64. doi: 10.1074/jbc.M304810200. Epub 2003 Jun 6. PMID: 12794070.)

Bakovic et al. also demonstrated that Sp1 binding site 3 (-148/-128) and 1 (-31/-9) might activate or repress transcription of CCTα depends upon the cellular background and Sp1 binding site 2 (-88/-50) would confer a positive regulation independent of the cell context. (Bakovic M, Waite KA, Vance DE. Functional significance of Sp1, Sp2, and Sp3 transcription factors in regulation of the murine CTP:phosphocholine cytidylyltransferase alpha promoter. J Lipid Res. 2000 Apr;41(4):583-94. PMID: 10744779.) 

2.We have supplemented the data confirming Sp1 overexpression, as shown in figure 4 (D, E).

3.In the study, we have demonstrated that glutamate down-regulates CCTα expression by specifically activating NMDA receptor, then inhibiting PC synthesis in lung tissues. Next, we further studied the mechanism of CCTα expression downregulation at the cellular level in A549 cells. In cell experiments, we directly treated A549 cells with NMDA, an NMDAR-specific agonist. Glutamate can activate other receptors in addition to NMDA receptor, so instead of treating A549 cells with glutamate, we used the specific agonist NMDA. 

The rescue experiments have been performed using the NMDAR blocker (MK-801) (Figure 7). 

4.We have added experiments to illustrate this point. the protein levels of Sp1 in the cytoplasm and nucleus of A549 cells after NMDA treatment were detected by Western blot (Figure 7A, B). As shown in figure 7A-D, The expression of Sp1 is increased in the cytoplasmic proteins and decreased in the nuclear proteins after treated with 300 μM NMDA (P<0.05, P<0.01) (Figure 7C, D). However, 50 μM MK-801 could reverse the increase of Sp1 expression in the cytoplasmic proteins and the decrease in the nuclear proteins induced by NMDA (P<0.05, P<0.01) (Figure 7C, D).

5.In figure 7E, the immunofluorescence staining of Sp1 after NMDA treatment in A549 cells has been re-detected and the Sp1 immunofluorescence data of NMDA treatment has been re-collected. The quality of image data has been improved. We use a magnification of 20×. Higher magnification images are very blurry.

6.In figure 2B, the “Glutamae” has been modified to “Glutamate”.

---

## [Decision Letter · Decision Letter 1]

22 May 2023

PONE-D-22-33593R1Sp1 mediated the inhibitory effect of glutamate on pulmonary surfactant synthesisPLOS ONE

Dear Dr. Luo,

Thank you for submitting your manuscript to PLOS ONE. After careful consideration, we feel that it has merit but does not fully meet PLOS ONE’s publication criteria as it currently stands. Therefore, we invite you to submit a revised version of the manuscript that addresses the points raised during the review process.

ACADEMIC EDITOR: Please carefully address the comments. Please be aware this is the final chance to revise the manuscript==============================

We look forward to receiving your revised manuscript.

Kind regards,

Yi Cao

Academic Editor

PLOS ONE

Reviewers' comments:

Reviewer's Responses to Questions

**Comments to the Author**

1. If the authors have adequately addressed your comments raised in a previous round of review and you feel that this manuscript is now acceptable for publication, you may indicate that here to bypass the “Comments to the Author” section, enter your conflict of interest statement in the “Confidential to Editor” section, and submit your "Accept" recommendation.

Reviewer #1: (No Response)

2. Is the manuscript technically sound, and do the data support the conclusions?

Reviewer #1: Partly

3. Has the statistical analysis been performed appropriately and rigorously? 

Reviewer #1: Yes

4. Have the authors made all data underlying the findings in their manuscript fully available?

Reviewer #1: Yes

5. Is the manuscript presented in an intelligible fashion and written in standard English?

Reviewer #1: Yes

6. Review Comments to the Author

Reviewer #1: Although the authors have responded to most of my previous comments and revised the manuscript accordingly, the data quality issues remain.

1. In Figures 4D and 4E, the Western blotting data can not correspond to quantified data.

2. In Figures 7B and 7D, which internal control is used for quantitative analysis? Although β-actin is also present in the nucleus, it is not suitable to be used as a control for nuclear protein samples.

3. In Figure 7E, the image data is highly blurry, which can not support their statement of Sp1 translocation.

7. PLOS authors have the option to publish the peer review history of their article (what does this mean?). If published, this will include your full peer review and any attached files.

Reviewer #1: No

---

## [Author Response · Author response to Decision Letter 1]

23 Jun 2023

Dear editor and reviewer,

Thank you very much for your letter and valuable comments and suggestions for our manuscript. We have modified and improved our manuscript according to your kind advice and the comments raised by the reviewers, and the amendments are highlighted in red in the revised manuscript. We sincerely hope that the revision is acceptable to be published on PLOS ONE. 

Below, please find the comments in black, followed by our responses in red. Exact changes in the manuscript are also presented in red font. 

Thank you very much for all your help and looking forward to hearing from you soon.

With best wishes,

Sincerely yours,

First author: Xiao-Hong Li

Corresponding authors: Zi-Qiang Luo and Chen Li, Email: luoziqiang@csu.edu.cn and Chen.physiology@outlook.com

Reviewer #1: Although the authors have responded to most of my previous comments and revised the manuscript accordingly, the data quality issues remain.

Responses: Thank you for your reviews.

1. In Figures 4D and 4E, the Western blotting data can not correspond to quantified data.

Response: Although the trend of band changes is not obvious when viewed with the naked eye, after statistical analysis, there are statistical differences between different groups. The analysis data of the bar chart is shown in the tables below. The analysis software we used is Bio Rad's Image Lab software. The bar chart and statistical analysis were performed using Image-Pro Plus 6.0 software.

2. In Figures 7B and 7D, which internal control is used for quantitative analysis? Although β-actin is also present in the nucleus, it is not suitable to be used as a control for nuclear protein samples.

Response: We have used the rabbit monoclonal antibody to Lamin A + Lamin C (ab108595), which is the nuclear envelope marker. The strip diagram of LaminA+C is shown in the figure 7C. In figures 7B, the GAPDH is used for quantitative analysis in the cytoplasmic proteins. In figures 7D, the Lamin A+C is used for quantitative analysis in the nuclear proteins. They are reflected in the annotation of the vertical coordinates in Figures 7B and 7D.

3. In Figure 7E, the image data is highly blurry, which can not support their statement of Sp1 translocation.

Response: In figure 7E, the image data we uploaded to the submission system is clear. The image data in the PDF of the manuscript is relatively blurry, possibly due to reduced pixels. We suggest that you download the original images in the review system to view it, which may be better. In figure 7E, the green color is Sp1, while the blue color is the nucleus. The green color is Sp1, while the blue color is the nucleus. Compared to the control group, the NMDA treatment group showed a decrease in overlapping green and blue cells, while MK-801 could alleviate the reduction caused by NMDA treatment. This indicates a decrease in Sp1 in the nucleus after NMDA treatment.

---

## [Decision Letter · Decision Letter 2]

4 Jul 2023

PONE-D-22-33593R2Sp1 mediated the inhibitory effect of glutamate on pulmonary surfactant synthesisPLOS ONE

Dear Dr. Luo,

Thank you for submitting your manuscript to PLOS ONE. After careful consideration, we feel that it has merit but does not fully meet PLOS ONE’s publication criteria as it currently stands. Therefore, we invite you to submit a revised version of the manuscript that addresses the points raised during the review process. Please submit your revised manuscript by Aug 18 2023 11:59PM. If you will need more time than this to complete your revisions, please reply to this message or contact the journal office at plosone@plos.org. Please include the following items when submitting your revised manuscript:A rebuttal letter that responds to each point raised by the academic editor and reviewer(s). You should upload this letter as a separate file labeled 'Response to Reviewers'.A marked-up copy of your manuscript that highlights changes made to the original version. You should upload this as a separate file labeled 'Revised Manuscript with Track Changes'.An unmarked version of your revised paper without tracked changes. You should upload this as a separate file labeled 'Manuscript'.

We look forward to receiving your revised manuscript.

Kind regards,

Yi Cao

Academic Editor

PLOS ONE

**Additional Editor Comments:**

Please carefully address the comments from the reviewer, and inadequate responses to the comments may lead to the rejection

Reviewers' comments:

Reviewer's Responses to Questions

**Comments to the Author**

1. If the authors have adequately addressed your comments raised in a previous round of review and you feel that this manuscript is now acceptable for publication, you may indicate that here to bypass the “Comments to the Author” section, enter your conflict of interest statement in the “Confidential to Editor” section, and submit your "Accept" recommendation.

Reviewer #1: (No Response)

2. Is the manuscript technically sound, and do the data support the conclusions?

Reviewer #1: Partly

3. Has the statistical analysis been performed appropriately and rigorously? 

Reviewer #1: Yes

4. Have the authors made all data underlying the findings in their manuscript fully available?

Reviewer #1: No

5. Is the manuscript presented in an intelligible fashion and written in standard English?

Reviewer #1: Yes

6. Review Comments to the Author

Reviewer #1: The authors claim that the NMDA treatment group showed decreased colocalization of Sp1 and nucleus, while MK-801 could alleviate the reduction caused by NMDA treatment. But they choose inappropriate data that may lead readers to confusion and misunderstanding. Because one cell expresses a highly green fluorescence signal, the field intensities lead to saturation in a few milliseconds, resulting in the other cells exhibiting a lower fluorescence signal. The author might explain that the expression of Sp1 was decreased under NMDA treatment, but this figure aims to illustrate the finding of nuclear-cytosol translocation of Sp1. They should provide three images representing blue (DAPI), green (Sp1), or yellow (merge) fluorescent channels, just like in the first version of this article, and quantify the percentage of cells with nuclear and cytoplasmic Sp1.

Although the result of nuclear and cytoplasmic protein extraction could support their finding (Effects of NMDA on Sp1 nuclear translocation in A549 cells), the immunofluorescence staining data are not.

7. PLOS authors have the option to publish the peer review history of their article (what does this mean?). If published, this will include your full peer review and any attached files.

Reviewer #1: No

---

## [Author Response · Author response to Decision Letter 2]

12 Jul 2023

Response: Thank you very much for the valuable feedback from the reviewer. The opinions of the reviewer are very important and reasonable. The Western blot results of nuclear and cytoplasmic protein extraction can totally support our finding that NMDA inhibits Sp1 nuclear translocation in A549 cells. The results of immunofluorescence staining are not very suitable. In order to avoid misleading the readers, we have decided not to present this result in the article. This does not affect our research conclusion (Effects of NMDA on Sp1 nuclear translocation in A549 cells). We have removed the contents about Sp1 fluorescence staining in the article.

---

## [Decision Letter · Decision Letter 3]

21 Jul 2023

Sp1 mediated the inhibitory effect of glutamate on pulmonary surfactant synthesis

PONE-D-22-33593R3

Dear Dr. Luo,

We’re pleased to inform you that your manuscript has been judged scientifically suitable for publication and will be formally accepted for publication once it meets all outstanding technical requirements.

Kind regards,

Yi Cao

Academic Editor

PLOS ONE

Additional Editor Comments (optional):

Reviewers' comments:

Reviewer's Responses to Questions

**Comments to the Author**

1. If the authors have adequately addressed your comments raised in a previous round of review and you feel that this manuscript is now acceptable for publication, you may indicate that here to bypass the “Comments to the Author” section, enter your conflict of interest statement in the “Confidential to Editor” section, and submit your "Accept" recommendation.

Reviewer #1: All comments have been addressed

2. Is the manuscript technically sound, and do the data support the conclusions?

Reviewer #1: Yes

3. Has the statistical analysis been performed appropriately and rigorously? 

Reviewer #1: Yes

4. Have the authors made all data underlying the findings in their manuscript fully available?

Reviewer #1: Yes

5. Is the manuscript presented in an intelligible fashion and written in standard English?

Reviewer #1: Yes

6. Review Comments to the Author

Reviewer #1: (No Response)

7. PLOS authors have the option to publish the peer review history of their article (what does this mean?). If published, this will include your full peer review and any attached files.

Reviewer #1: No

---

## [Editor Report · Acceptance letter]

31 Jul 2023

PONE-D-22-33593R3 

Sp1 mediated the inhibitory effect of glutamate on pulmonary surfactant synthesis 

Dear Dr. Luo:

I'm pleased to inform you that your manuscript has been deemed suitable for publication in PLOS ONE. Congratulations! Your manuscript is now with our production department. 

Kind regards, 

on behalf of

Dr. Yi Cao 

Academic Editor

PLOS ONE